# Convolutional Neural Networks Based on Sequential Spike Predict the High Human Adaptation of SARS-CoV-2 Omicron Variants

**DOI:** 10.3390/v14051072

**Published:** 2022-05-17

**Authors:** Bei-Guang Nan, Sen Zhang, Yu-Chang Li, Xiao-Ping Kang, Yue-Hong Chen, Lin Li, Tao Jiang, Jing Li

**Affiliations:** State Key Laboratory of Pathogen and Biosecurity, Beijing Institute of Microbiology and Epidemiology, Academy of Milltary Medical Sciences, Beijing 100071, China; nbg2012@163.com (B.-G.N.); zhangsen2260@163.com (S.Z.); liyuchang66@163.com (Y.-C.L.); kangxiaoping@163.com (X.-P.K.); happy@163.com (Y.-H.C.); dearwood@sina.com (L.L.)

**Keywords:** sequential amino acid frequency, deep learning, adaptation, SARS-CoV-2, Omicron

## Abstract

The COVID-19 pandemic has frequently produced more highly transmissible SARS-CoV-2 variants, such as Omicron, which has produced sublineages. It is a challenge to tell apart high-risk Omicron sublineages and other lineages of SARS-CoV-2 variants. We aimed to build a fine-grained deep learning (DL) model to assess SARS-CoV-2 transmissibility, updating our former coarse-grained model, with the training/validating data of early-stage SARS-CoV-2 variants and based on sequential Spike samples. Sequential amino acid (AA) frequency was decomposed into serially and slidingly windowed fragments in Spike. Unsupervised machine learning approaches were performed to observe the distribution in sequential AA frequency and then a supervised Convolutional Neural Network (CNN) was built with three adaptation labels to predict the human adaptation of Omicron variants in sublineages. Results indicated clear inter-lineage separation and intra-lineage clustering for SARS-CoV-2 variants in the decomposed sequential AAs. Accurate classification by the predictor was validated for the variants with different adaptations. Higher adaptation for the BA.2 sublineage and middle-level adaptation for the BA.1/BA.1.1 sublineages were predicted for Omicron variants. Summarily, the Omicron BA.2 sublineage is more adaptive than BA.1/BA.1.1 and has spread more rapidly, particularly in Europe. The fine-grained adaptation DL model works well for the timely assessment of the transmissibility of SARS-CoV-2 variants, facilitating the control of emerging SARS-CoV-2 variants.

## 1. Introduction

The sustained coronavirus disease 2019 (COVID-19) pandemic has caused more than 500 million infection cases and over 6 million deaths [1]. The huge infection population provides an ideal breeding ground to accumulate genomic mutations and to select more adaptive pandemic strains for severe acute respiratory coronavirus 2 (SARS-CoV-2), the causative agent of COVID-19 [2]. Variants with a higher and higher ability to spread have been emerging, thus exacerbating the COVID-19 pandemic with a mechanism of positive feedback. Variants of concern (VOCs), named by the World Health Organization, have taken turns dominating the SARS-CoV-2 population due to their higher transmissibility compared to their ancestral strains, starting with the emergence of the first VOC, the Alpha (B.1.1.7) variant, in January 2021 in the UK [3]. In particular, the latest VOC, Omicron (B.1.1.529), first recorded in South Africa on 24 November 2021 [4], showed an average basic reproductive number (R0) of 8.2 [5], much higher than the R0 of 5.08 [6] of the former dominating VOC, Delta (B.1.617.2). The Omicron VOC causes an asymptomatic or mildly symptomatic infection [7,8], stimulates a weaker immune response [9], and fewer responses to vaccine-promoted SARS-CoV-2 neutralizing antibodies [10]. The Omicron spike protein also shows an enhanced binding affinity to the host ACE2 receptor compared with other variants [11,12]. All the above-mentioned challenge the control and prevention strategies against SARS-CoV-2 in any country, including China [13]. Therefore, the most effective approach to prevent and control the COVID-19 pandemic is to determine candidate variants for VOCs and to make additional efforts to control their spread.

WHO has defined variants of concern (VOCs) based on reliable evidence of increased phenotypes, including transmissibility (https://www.who.int/en/activities/tracking-SARS-CoV-2-variants/, accessed on 28 March 2022). Biological approaches have determined the significance of key SARS-CoV-2 VOC mutations that promote transmission, enhance receptor binding affinity [14,15], and antagonize immunity [16,17], however, which lag behind a timely concern about them. The Omicron VOC (B.1.1.529) has dominated the COVID-19 pandemic, spreading worldwide shortly after emerging in South Africa in November 2021 [18]. Omicron has accumulated over 50 mutations in its genome, with over 30 mutations residing in the spike protein [18]. Such a large pool of variants of Omicron lineage has posed a high risk of producing more transmissible variants. Therefore, it is urgent to predict in real-time the risk of SARS-CoV-2 variants, particularly high-risk Omicron variants, independent of epidemiological evidence and experimental studies.

Human adaptation is a prerequisite for causing infection in the population or even a sustained pandemic, for influenza [19,20] or coronaviruses [21,22]. Thus, it is an alternative risk assessment strategy to assess the human adaptation of any SARS-CoV-2 variant. Machine learning (ML) and deep learning (DL) have performed well to predict human adaptation and virus evolution based on the nucleotide composition of viral genomes [19,23]. A discriminable compositional dinucleotide (DNT) of CpG has been observed in the SARS-CoV-2 genome [24]. However, nucleotide compositional analysis poses the concern of too much dimension reduction, losing genomic sequential information. In our previous study, we built a DL model to predict the human adaptation of SARS-CoV-2 with two types of adaptation labels based on the DNT cluster, DNT representation (DCR). A higher adaptation of Delta and Omicron was predicted [21]. However, the model was based on the training data of coronaviruses from various types of animal hosts, without SARS-CoV-2, and thus was coarse-grained to predict the sublineages of Omicron.

In the present study, we decomposed sequential amino acid (AA) distribution in the Spike-coding region of SARS-CoV-2 viruses. Unsupervised machine learning methods of t-distributed stochastic neighbor embedding (t-SNE), Principal Component Analysis (PCA), and hierarchical clustering were performed to observe the AA distribution in Spike among the SARS-CoV-2 lineages of Alpha, Delta, and others. Then, a Convolutional Neural Network (CNN) model was built with three types of human adaptation labels to predict the adaptation of SARS-CoV-2 Omicron sublineages. Our model provides a valuable model for the real-time assessment of the risk of any emerging variant of SARS-CoV-2 Omicron sublineages and others.

## 2. Methods

### 2.1. Preparation of Spike Sequences and Representation of Sequential Amino Acid (AA)

More than 9,000,000 full genome sequences of SARS-CoV-2 viruses were downloaded from the Global Initiative on Sharing All Influenza Data (GISAID) database [25], up to 28 March 2022. An amount of 4,080,132 high-quality Spike sequences, with degenerate and uncertain nucleotides (nts) less than 0.005% and a further 3,833,513 Spike sequences with degenerate and uncertain nucleotides less than 0.001%, were selected and aligned with a uniform sequence length of 3822 nts. Approximately 4% (160,000) of the high-quality Spike sequences were comprehensively sampled for this study, considering the sample coverage and calculation consumption (with memory occupation less than 500 GB). Sequence data were shuffled and then were randomly sampled with a python package, pandas.sample (random_state = 1), to select 20,000 Spike sequences, respectively, for Iota and Epsilon SARS-CoV-2 variants, and 40,000 sequences, respectively, for Gamma, Alpha, Delta, and Omicron SARS-CoV-2 variants. The metadata and date/location files were also downloaded and added for each of the sampled sequences. Sequence ID and lineage information for all sampled SARS-CoV-2 viruses are recorded in Appendix A. The detailed information, including collection date, location, and others of the sampled 40,000 Omicron sequences are listed in Appendix A.

Each Spike sample was translated into a protein sequence and then sliced into serial fragments, with the same length of 64 AAs and one AA shifted forward for any two neighboring fragmented sequences, using the sliding window method (window size = 64 AAs, stride = 1 AA). The first fragment was spliced with the last 63 AAs at the ending terminal of the sequence and the first AA at the starting terminal of the sequence. The compositional AAs for each of the 20 types of AAs were calculated as a frequency value total for each fragment (Formula I), and thus each AA in the Spike protein sequence was represented as a frequency vector of 20 types of AAs of the fragment.


**Formula I:**

(1)
$$freqx=\frac{\Sigma x}{\sum_{i=1}^{20}x},\;x\;\in \left[I,\;D,\;M,\;\ldots ,\;\;\;\;\;\;\;F,\;N,\;K\right]$$



### 2.2. Unsupervised Machine Learning (ML)

Sklearn.decomposition.PCA was utilized for Principal Component Analysis (PCA) [26], with which the nucleotide composition features for multiple segments were reduced into one principal component with the largest possible variance (Formula II). Another unsupervised ML approach, hierarchical clustering, was utilized for the hierarchical cluster analysis of Spike sequences. Spikes were clustered into various hierarchical groups based on the Euclidean distance in nucleotide compositional values, according to Formula III. PCA and t-SNE were performed to observe the distribution of compositional traits. Two components (PCA1 and PCA2 or t-SNE1 and t-SNE2) were reduced from a feature matrix of 1273 × 20 for each Spike sequence.


**Formula II:**

(2)
$$minimize\;{\left\|A-XY\right\|}_{F}^{2}=\overset{m}{\underset{i=1}{\sum }}\{\overset{n}{\underset{j=1}{\sum }}{\left(A_{ij}-x_{i}y_{j}\right)}^{2}\;\},\;s.t.\;\;X\epsilon R^{m\times k},\;Y\epsilon R^{k\times n},\;k<m\;or\;n$$




**Formula III:**

(3)
$${\left\|a-b\right\|}_{2}=\sqrt{\sum_{i}{\left(a_{i}-bi\right)}^{2}},\;\;a,\;b=AA\;vector\;for\;site\;i,\;\;i\;\in \left(1,\;1273\right)$$



To compare the difference in AA frequency vectors at each of the 1273 AA sites in Spike, for any two groups of the sequences of adaptation 0, adaptation 1, adaptation 2, and Omicron, dot product was performed for two AA frequency vectors at each site. the product value was negatively correlated with the different degrees of AA frequency.
(4)Dot (a, b)=a1 b1+a2 b2+···+a20 b20, a, b=AA frequency

### 2.3. Model Architecture of a 2D-CNN

Gamma, Alpha, and Delta SARS-CoV-2 variants were labeled with three types of human adaptation (0, 1, and 2, respectively, for lower, middle-level, and higher adaptation), and 20,000 samples for each adaptation type were randomly sampled from the above-mentioned dataset to build a training/validating dataset of 60,000 Spike sequences. A data split was performed post a data shuffle with a test size of 0.2 via the train_test_split model in sklearn.model_selection. Spike samples were subjected to adaptation prediction with the trained 2D-CNN model. Both the training and validation data were reshaped into a 159 × 160 array with one channel from the array of 1272 × 20, with the first AA vector of 20 values abandoned. The 2D-CNN was constructed with four layers of convolution calculations with 8, 16, 32, and 64 out channels, with a stride of 1, and a padding of 1. The array of 159 × 160 was convoluted with a ReLU function (Formula IV) into a flattened vector of 5760, which was activated with a sigmoid function (Formula V) and reduced to a vector of 160, which was linearized into three predicted values. Adaptation was predicted according to the maximum value of the predicting vector of three values (respectively for adaptations 0, 1, and 2) for the three above-mentioned adaptation types. To validate the 2D-CNN performance, confusion matrices and micro-average ROC curves with AUCs were drawn [26,27].


**Formula IV:**

(5)
$$\mathrm{ReLU}\;\mathrm{function}:\;f\left(x\right)=\max(0,\;w^{T}x+b)$$




**Formula V:**

(6)
$$\mathrm{Sigmoid}\;\mathrm{function}:\;f\;\left(x\right)=1\;/\;\left(1+e^{-x\;}\right),$$



### 2.4. Supplementary Data and Code Availability

Original sequence data (up to 28 March 2022) were available in the Global Initiative on Sharing All Influenza Data (GISAID) database [3]. All original and processed data for the results of the present study were available upon request. Scripts for the project were available on Github: https://github.com/Jamalijama/OmicronAdaptation, accessed on 18 April 2022.

### 2.5. Data Analysis or Statistical Analysis

Random sampling was used to obtain the sequences in the training and validation datasets. The unsupervised ML models, PCA, and t-SNE were utilized to observe the distribution of compositional traits. Reduced PCA (PCA1 and PCA2) or t-SNE (t-SNE1 and t-SNE2) values were visualized with the Python-Seaborn model. The performance of the 2D-CNN model was validated through confusion matrices and micro-average ROC curves with AUCs. A dot product was calculated to compare the AA distribution at each site for the different adaptation groups. Statistical significance in the PCA1 value difference was analyzed according to an unpaired, nonparametric Mann–Whitney test, with a hypothesis of a non-Gaussian data distribution, with Graphpad Prism 9.

## 3. Results

### 3.1. Methodology of a Deep Learning Model to Predict Human Adaptation of SARS-CoV-2 Variants

As the architecture diagram (Figure 1) shows, data downloading and cleaning were performed for all types of SARS-CoV-2 variants, and only sequences with uncertain or degenerate base ratios of less than 0.001 were retained (Figure 1A). Then, the Spike genes were aligned to keep a uniform CDS sequence length of 3822 and further decomposed as a frequency value of each of 20 amino acids to analyze the genomic characteristics, as shown in Figure 1B. All SARS-CoV-2 Spike gene sequences, except Omicron variants, were randomly divided into the training and validation datasets to build a human adaptability classification model, which would be utilized to predict the Omicron variant’s human adaptation. A 2D-CNN framework for three-category adaptation classification, which was higher adaptation (2), middle adaptation (1), and lower adaptation (0), was built with four layers of convolution + ReLU, three average pooling layers + one maximum pooling layer, three fully connected layers, and a final softmax layer, to predict the human adaptation of each Omicron variant (Figure 1C). The unsupervised projection methods of t-distributed stochastic neighbor embedding (t-SNE) and Principal Component Analysis (PCA) were utilized to learn the separation of different types of VOCs/VOIs (Figure 1D). Temporal and spatial shifts of Omicron variants’ adaptation were further analyzed (Figure 1E), and AA traits important for different human adaptations of SARS-CoV-2 variants were assessed by dot product in Spike genes (Figure 1F).

### 3.2. Genomic Codons Discriminate the Difference between SARS-CoV-2 Variants

To evaluate the possibility of predicting human adaptation based on amino acid composition characteristics, firstly we analyzed the general separability of AA traits in SARS-CoV-2 S genes. The two-dimensional t-SNE or PCA projection of the AA features of a total of 6000 sampled S sequences randomly selected from Alpha, Delta, Omicron, Gamma, Iota, and Epsilon variants (1000 for each) indicated an obvious separation between Alpha, Delta, Omicron, and other VOCs/VOIs in the two reduced t-SNE components (Figure 2A). The data linearity was further assessed to reflect its continuity and differentiability and to support the machine/deep learning classification of these samples. The linearity feature was designed as the ratio of the data range of PCA1 to the data range of PCA2 based on the orthogonal distribution between PCA1 and PCA2 with Omicron variants (Figure 2B) or without them (Figure 2C). These results showed a clear separation of Alpha, Delta, Omicron, and other VOCs/VOIs; however, Omicron variants were dispersed at least in three clusters, which meant that they possess different characteristics of human adaptation. The hierarchical clustering diagram further proved the clear discrimination of Delta (high-adaptation), Alpha (middle-adaptation), and other VOCs/VOIs (low-adaptation) (Figure 2D). Taken together, the above-mentioned results indicate a general separability and linearity of the AA traits in different types of SARS-CoV-2 VOCs/VOIs, including Omicron variants.

### 3.3. Importance of Human Adaptation-Associated Codons in the Spike of SARS-CoV-2

A dot product was generated to analyze the differences in the AA composition among the types of human adaptation of SARS-CoV-2 in S genes. All of the significantly different AA sites were obtained with a normalized dot product value among all sites. Then, 20 consecutive amino acid sites were compared as a cluster to make the comparison more intuitive. It showed that dominantly divergent sites were located within the range of 80 to 320 AA sites between the variants with adaptation 0 and adaptation 1 (Figure 3A). There were, additionally, more site(s) around 500, with ranges of 700–760, 900, 960, 1260, and 1280 observed between the variants with adaptation 2 and adaptation 0 (Figure 3B), and an additional range of 1120–1180 sites was found between the variants with adaptation 2 and adaptation 1 (Figure 3C). When comparing Omicron variants with these three adaptation variants, almost all of the important AA sites mentioned above can be seen (Figure 3D−F). Thus, the importance of human adaptation-associated AAs in the Spike of SARS-CoV-2 can be determined well.

### 3.4. Performance of the Deep Learning Model to Predict SARS-CoV-2 Adaptation

To assess the ability of SARS-CoV-2 variants of Omicron sublineages to spread throughout a population, we built a CNN model based on the AA frequency vector at each AA site in Spike CDS, with training/validating data of Iota, Epsilon, Gamma (adaptation label of 0 for low adaptation), Alpha (adaptation label of 1 for a middling degree of adaptation), and Delta (adaptation label of 2 for high adaptation). To evaluate the training effect of CNN, we output the data of full-connected layers and PCA was performed to visualize the distribution of these data in the space with three adaptation labels. A lesser classification of the three groups of sequences was observed post 10 training epochs (Figure 4A), without a clear separation between adaptation 0 and 2 in PCA1, or post 20 epochs (Figure 4B), without separation between adaptation 1 and 2 in PCA2; the highest separation in both PCA1 and PCA2 between any two groups was obtained after training epoch 30 (Figure 4C). The training loss of the CNN model is also indicated (Figure 4D), with a quick and sharp drop for about three epochs. The high accuracy was also confirmed by the confusion matrix (Figure 4E) and ROC (Figure 4F) curve for the model after training epoch 30. Thus, the sequential AA frequency-based CNN is convergent with the training/validating data of SARS-CoV-2 lineages at the early COVID-19 stage and is valuable to predict the human adaptation of other SARS-CoV-2 variants.

### 3.5. Human Adaptation of SARS-CoV-2 Omicron Variants

The adaptation of all 40,000 Omicron variants was predicted based on the sequential AA frequency of Spike and was performed with the trained CNN predictor. A total of 70.60% (28,241/40,000) were predicted as middle-level adaptations (adaptation 1), 29.40% (11,758/40,000) were more highly adaptive to humans (adaptation 2), and only one sequence was predicted as having low adaptation (Table 1). For the 6317 Spike sequences from the test data of 40,000 sampled sequences in March 2020, 4822 samples (76.33%) were more highly adaptive. The temporal and spatial distribution of variants with adaptation 1 or adaptation 2 were curved, based on the collection date and location information (Figure 5). Omicron variants with middle-level adaptation surged since November 2021, peaked firstly in Africa in December 2021 (Figure 5C), then in Asia (Figure 5B), Europe (Figure 5D), and North/South America (Figure 5E,F) in January 2022, and have declined worldwide up to now. However, there have been variants with high adaptation since December 2021, particularly in Europe and Africa (Figure 5C,D). Interestingly, such predicted adaptation labels of 1 and 2 were highly consistent with the sublineage classification of BA.1 and BA.2: 98.50% (27,986/28,412) of BA.1/BA.1.1 variants were predicted as adaptation 1, whereas 98.19% (11,289/11,497) of BA.2 variants were predicted as adaptation 2. Thus, the present CNN model provides a real-time alternative approach to assess human adaptation as a transmissibility index of emerging SARS-CoV-2 variants.

## 4. Discussion

The causality of genotype–phenotype was the decisive basis for every virus phenotype, such as transmissibility and pathogenicity. Adaptation is a valuable index to assess the transmissibility and pathogenicity of viruses [19,20,21,28,29] and is predictable based on viral compositional features of DNTs, DCR [21], codons, and AAs [23] in the genome. Our previous DCR-based 3D-CNN model could predict the type I and II human adaptations, high pathogenicity/low transmissibility, and low pathogenicity/high transmissibility, respectively, of SARS-CoV-2 Alpha and Delta VOCs [21], with the training/validating data of coronaviruses other than SARS-CoV-2. However, such a model was based on genomic compositional features and was not competent in discriminating the fine-grained sequential variance in virus genes. The approach in this study to represent genomic information was sequence-dependent and capable of discriminating any of the amino acid mutations in the virus gene. Accordingly, the represented sequential AA frequency was fine-tuned and sensitive enough to indicate the separation of VOIs/VOCs of SARS-CoV-2 lineages. Particularly, PCA results showed a significantly greater distance between Omicron variants and other VOIs/VOCs. The viral Spike protein is a receptor-binding protein that is sensitive to AA mutations at a specific site but not to the same mutation at other sites, indicating a strict local AA sequence dependence. Each AA was represented as an AA vector of the relative frequency of this AA to all AAs, and thus was more representative than the scalar AA information. The comparison of each pair of AA vectors at each site between any two SARS-CoV-2 lineages revealed the importance of each site to the difference between any two SARS-CoV-2 lineages. Our results indicated dominant divergent domains in the N terminal of Spike (80–320) between the variants with various adaptation labels, additionally with near 500, 760–780, 960, and around 1280 found for Omicron variants, different from other variants. Therefore, the total of 25,460 values for each Spike protein provided fine-tuned enough information to tell apart the tiny variances.

The sequential 25,440 (20 values for the first AA in Spike, methionine, were excluded) values were reshaped to a matrix of 159 × 160, which was utilized for the CNN modeling. The sequential AA frequency matrix-based CNN was quickly convergent after four convolutional layers. It accurately classified SARS-CoV-2 variants of Gamma, Alpha, and Delta, respectively, as the three adaptation types of 0 (lower), 1 (middle-level), and 2 (higher), on randomly sampled cross-validation data. The trained model predicted the accuracy of randomly split data of any of the above-mentioned variants. Surprisingly, 98.50% (27,986/28,412) of BA.1/BA.1.1 variants were predicted as adaptation 1, whereas 98.19% (11,289/11,497) of BA.2 variants were predicted as adaptation 2, implying a higher adaptation of BA.2 variants than BA.1 variants. There are limited epidemiological and experimental results to show the prevalence potential of Omicron sublineages. Omicron BA.2 has been indicated to have an “increased growth rate” in England compared to the BA.1 sublineage [30], has replaced BA.1 in Japan [31], and has shown a high potential for becoming the next dominant variant [32]. Thus, the CNN based on sequential *Spike* predicting the higher human adaptation of the BA.2 sublineage was consistent with pandemic facts. In the context of COVID-19 disease severity, post-SARS-CoV-2 infection, decreasing pathogenicity was reported by recent variants compared to others in the early stages of the COVID-19 pandemic [33]. However, there is not yet conclusive about the severity difference between Omicron BA.1 and BA.2 sublineages. The adaptation prediction of these variants implies less pathogenicity for BA.2 than BA.1.

There are lots of mutations among Omicron and other SARS-CoV-2 variants with respect to the N-terminal domain and the receptor-binding domain in Spike [34], as was also indicated by our sequence representation approach. The immune breakthrough by Omicron sublineages is controversial. Although vaccinated subjects may not be equally protected against all SARS-CoV-2 lineages [35], the vaccine is an effective means of preventing COVID-19 infection. Experimental results indicate that currently available vaccines provide robust protection against all three BA.1 and BA.2 Omicron sublineages [35], without a significant inter-sublineage difference. On the other hand, breakthrough infections with SARS-CoV-2 Omicron have also been observed in people after being boosted with the mRNA vaccine [36]. Therefore, it is hard to understand the expansion of BA.2 sublineage variants based on antibody evasion in the population. In consideration of the high importance of the human adaptation of coronaviruses [21] or influenza viruses [19], the human adaptation prediction of the Omicron sublineages in the present, or those possibly emerging in the future, is an alternative assessment for the prevalence potential of Omicron sublineages and other variants. The present CNN model is capable of assessing, in real-time, human adaptation as a transmissibility index of emerging variants of SARS-CoV-2. It is a valuable alternative step via predicting the adaptation to assess the transmission potential of any emerging SARS-CoV-2 variant in Omicron sublineages.

## 5. Conclusions

Summarily, the Omicron BA.2 sublineage is more adaptive than BA.1/BA.1.1 and has spread more rapidly, particularly in Europe. The fine-grained adaptation DL model works well for the timely assessment of the transmissibility of SARS-CoV-2 variants, facilitating the risk prediction and control of emerging SARS-CoV-2 variants.

## Figures and Tables

**Figure 1 viruses-14-01072-f001:**
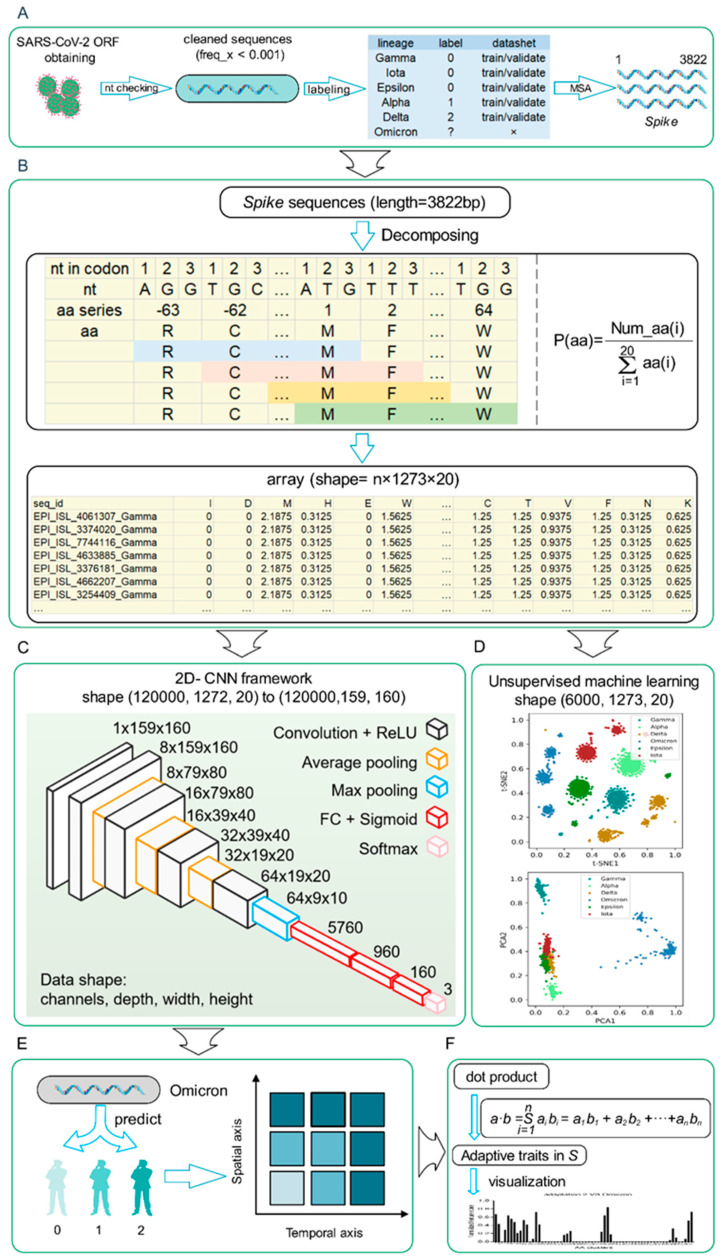
Workflow to decompose sequential amino acid (AA) frequency and to build the deep learning model for adaptation prediction. The workflow was designed from the top to bottom as follows: sequence cleaning and sampling of SARS-CoV-2 *Spike* genes (**A**), sequence decomposing diagram (**B**), design of the 2D-CNN model (**C**), unsupervised machine learning of randomly selected SARS-CoV-2 *Spike* genes (**D**), human adaptability prediction of Omicron variants (**E**), and the assessment of amino acid features for human adaptation (**F**).

**Figure 2 viruses-14-01072-f002:**
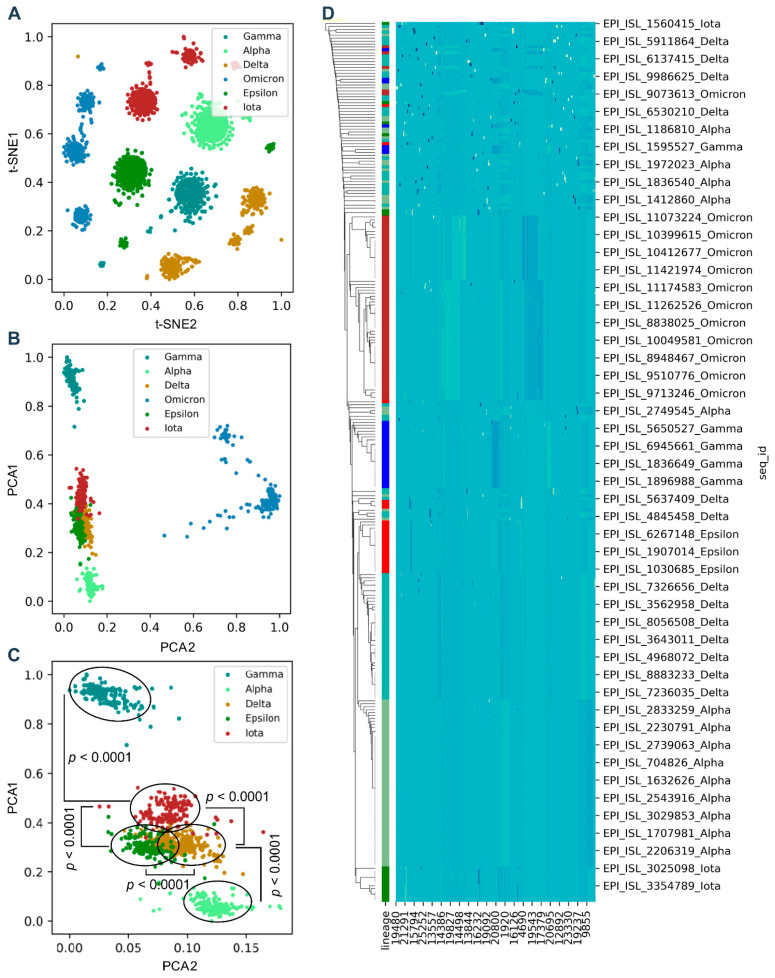
Projection of sequential AA frequency in *Spike* with t-SNE and PCA. Plot of two components reduced by t-SNE from 1273 × 20 features (**A**). The plot represents the PCA1/PCA2 ratio of the absolute difference value for truncated data between the 20% and 80% quantiles. PCA1 and PCA2 were affined in space with the data point of the least PCA1 value as the coordinate origin and were then calculated for simple linear regression for Spike genes with (**B**) or without (**C**) Omicron variants, respectively. The hierarchical clustering diagram of Alpha, Delta, Omicron, and other VOCs/VOIs (**D**). Statistical significance in the PCA1 value difference between two neighboring VOCs/VOIs is indicated, respectively, according to an unpaired, nonparametric Mann–Whitney test.

**Figure 3 viruses-14-01072-f003:**
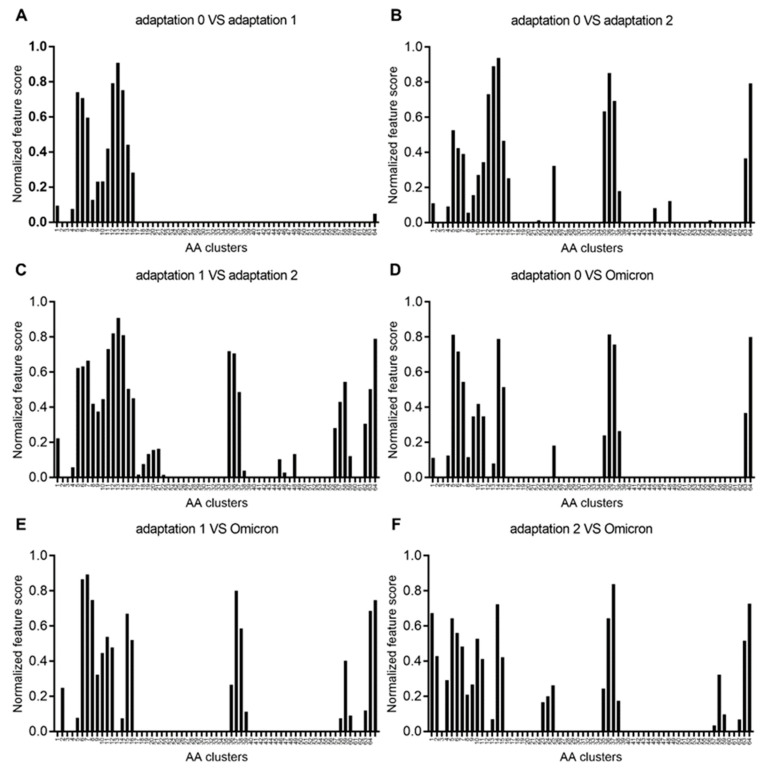
Importance of serially segmented AA clusters for three types of human adaptation of SARS-CoV-2 variants. Normalized dot product value of AA sites in Spike genes between adaptation 0 and adaptation 1 (**A**), adaptation 0 and adaptation 2 (**B**), adaptation 1 and adaptation 2 (**C**), adaptation 0 and Omicron (**D**), adaptation 1 and Omicron (**E**), and adaptation 2 and Omicron (**F**).

**Figure 4 viruses-14-01072-f004:**
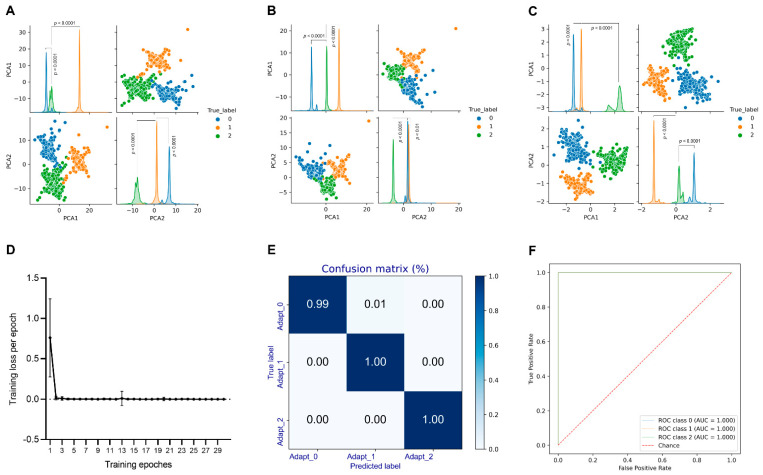
Performance of sequential AA-based CNN to predict the human adaptation of SARS-CoV-2 variants. (**A**–**C**) The performance of the sequential AA-based CNN model was assessed by the pair plot of PCA1 and PCA2 of the full connection layers from the CNN model post-training epochs of 10, 20, and 30 ((**A**–**C**) for training epochs of 10, 20, and 30, respectively); (**D**) The training loss for the CNN model was curved for 30 training epochs; (**E**,**F**) Confusion matrix (**E**), receiver operating characteristic curve (ROC) (**F**), and training ((**D**–**F**) for training epochs of 30). Statistical significance in the PCA1 or PCA2 values of the full connection layers between two neighboring adaptation groups is indicated, respectively, according to an unpaired, nonparametric Mann–Whitney test.

**Figure 5 viruses-14-01072-f005:**
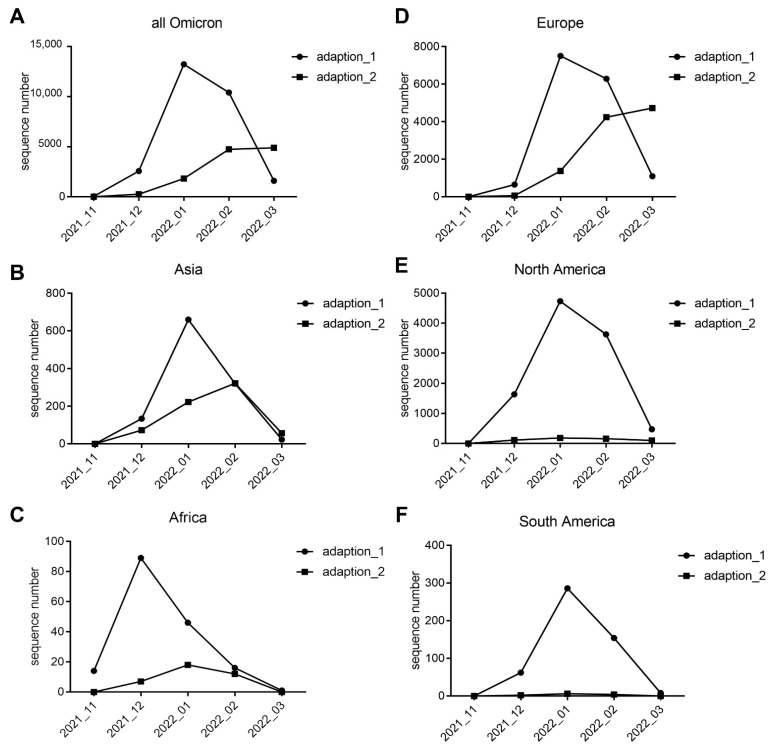
Prediction and prediction probability of Omicron sublineages. (**A**) Temporal analysis of Omicron variants which were predicted for their human adaptation (adaptation_1 and 2, the only 1 sample with adaptation 0 was excluded); (**B**–**F**) Spatial distribution of the Omicron variants with the predicted adaptation, 1 or 2, in Asia (**B**), Africa (**C**), Europe (**D**), North America (**E**), and South America (**F**).

**Table 1 viruses-14-01072-t001:** Adaptation prediction of sampled Omicron variants of various sublineages, with the trained CNN model.

Sublineages	Total	Adaptation 0	Adaptation 1	Adaptation 2
BA.1	28,412	1	27,986	425
BA.2	11,497	0	208	11,289
BA.3	10	0	10	0
Nan	81	0	37	44
Total	40,000	1	28,241	11,758

## Data Availability

Original sequence data (up to 28 March 2022) were available in the Global Initiative on Sharing All Influenza Data (GISAID) database. All original and processed data for the results of the present study were available upon request. Scripts for the project were available on Github: https://github.com/Jamalijama/OmicronAdaptation, accessed on 18 April 2022.

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
