# Peer review of "Convolutional Neural Networks Based on Sequential Spike Predict the High Human Adaptation of SARS-CoV-2 Omicron Variants"

_viruses, 2022, doi:10.3390/v14051072_

Round 1

Reviewer 1 Report

• The primary output/endpoint variable(s)/measurements of the study should be defined. 

• What are the inclusion and exclusion criteria in the study?  • Which randomization method was used in the distribution of the individuals included in the study to the groups?  • Which blinding (masking) method was used in the study?  • Data analysis or Statistical analysis sub-section title should be added to the Materials and Methods.  • How was the sample size determined? This information should be explained in the Materials and Methods section.  • Which sampling (probable or non-probable, etc.) method was used in the study?  • Statistical tests for hypothesis testing and their assumptions should be specified in the statistical analysis of the study in the Materials and Methods section.  • The details (version, license number, etc.) of the statistical package(s) or program(s) should be given in the section of "Data Analysis or Statistical Analysis".  • It should be explained how the qualitative and quantitative data are summarized under the sub-heading of Statistical Analyzes in the Materials and Methods section of the study.  • The exact P values should be added to the table(s) (p=0.25; p=0.03).  • Which methods are used to model relationships between variables?  • The descriptions and other descriptive values/data should be defined on the tables and shapes. • Are the data subjected to pre-processing?  • How were extreme/outlier values in the data determined and resolved?  • The number of current references on the subject of the study should be increased.  • The discussion section of the research can be expanded by supporting current studies to address the findings of other studies reported with the present findings.  • What approaches were used to test the validity of the models? • Which metrics were used in the performance evaluation of the estimates of models/algorithms?   • Which method(s) was/were used to optimize the hyperparameters of models/algorithms?

Author Response

The primary output/endpoint variable(s)/measurements of the study should be defined. 

Response: Thank you. As depicted in Method section, variables of compositional amino acids were dimensionless, there were represented as the frequency of each amino acid to the total of 20 amino acids.

What are the inclusion and exclusion criteria in the study? 

Response: Thank you. As depicted in Method section, all SARS-CoV-2 sequences up to March 28th, 2022, were downloaded. However, the sequences with low quality were excluded, and the included sequences were randomly sampled with python package of pandas.sample (random_state = 1).

Which randomization method was used in the distribution of the individuals included in the study to the groups? 

Response: As depicted in Method section, randomization was performed with python package of pandas.sample (random_state = 1).

Which blinding (masking) method was used in the study? 

Response: Blinding (masking) method was not applicable and not utilized in this study.

Data analysis or Statistical analysis sub-section title should be added to the Materials and Methods. 

Response: Thank you. Data analysis or Statistical analysis sub-section title has been added.

How was the sample size determined? This information should be explained in the Materials and Methods section. 

Response: Thank you. We have added the sample size determination information in the Materials and Methods section: comprehensively considering the sample coverage and calculation consumption.

Which sampling (probable or non-probable, etc.) method was used in the study? 

Response: As depicted in Method section, randomization was performed with python package of pandas.sample (random_state = 1).

Statistical tests for hypothesis testing and their assumptions should be specified in the statistical analysis of the study in the Materials and Methods section. 

Response: Thank you. We have specified. Statistical significance in the PCA1 value difference were analyzed according to unpaired, nonparametric Mann-Whitney test, with a hypothesis of non-Gaussian data distribution.

The details (version, license number, etc.) of the statistical package(s) or program(s) should be given in the section of "Data Analysis or Statistical Analysis". 

Response: Thank you. We have added the software version information:  Graphpad Prism 9.

It should be explained how the qualitative and quantitative data are summarized under the sub-heading of Statistical Analyzes in the Materials and Methods section of the study. 

Response: Thank you. We have added such description accordingly.

The exact P values should be added to the table(s) (p=0.25; p=0.03). 

Response: Thank you. It was not applicable.

Which methods are used to model relationships between variables? 

Response: As depicted in Materials and methods section, Convolutional neural network.

The descriptions and other descriptive values/data should be defined on the tables and shapes.

Response: Thank you. We have added such description, about how the data was out-put.

Are the data subjected to pre-processing? 

Response: Thank you. No, sequence-to-prediction.

How were extreme/outlier values in the data determined and resolved? 

Response: Thank you. The sequence inclusion / exclusion performance had guaranteed the data quality. There were no extreme/outlier values in the data.

The number of current references on the subject of the study should be increased. 

Response: Thank you. We have added more references on the subject of the study.

The discussion section of the research can be expanded by supporting current studies to address the findings of other studies reported with the present findings. 

Response: Thank you. We have revised the discussion section extensively.

What approaches were used to test the validity of the models?

Response: Thank you. We have added more references on the subject of the study.

Which metrics were used in the performance evaluation of the estimates of models/algorithms? 

Response: Thank you. This study was the first one to predict the adaptation of SARS-CoV-2 Omicron variants based of other SARS-CoV-2 VOCs, there was no benchmark to evaluate the model/algorithms. However, the model performance was well evaluated with standard deep learning method of cross-validation test data.

Which method(s) was/were used to optimize the hyperparameters of models/algorithms?

Response: Thank you. As depicted in method section. The optimization was performed via balancing data volume, adding CNN layers, training with various training epochs, as was indicated in the available code: https://github.com/Jamalijama/OmicronAdaptation.

Reviewer 2 Report

This work presents a study on the adaptation prediction of Omicron variants. Omicron BA.2 sublineage was more adaptive than BA.1/BA.1.1 and 285 spreaded more rapidly, particularly in Europe. This work is informative, however some corrections/revision required as listed below:

  1. Editing required on the reference section. Many references are not complete and some recent work on omicron need to be cited, for examples https://doi.org/10.1016/j.compbiomed.2022.105367
  2. Conclusion section should be more comprehensive and some future steps to fight this variant may be added.
  3. How much dangerous these variants need to be discuss (fatal rate), even though the transimissibility is high.

Author Response

This work presents a study on the adaptation prediction of Omicron variants. Omicron BA.2 sublineage was more adaptive than BA.1/BA.1.1 and 285 spreaded more rapidly, particularly in Europe. This work is informative, however some corrections/revision required as listed below:

  1. Editing required on the reference section. Many references are not complete and some recent work on omicron need to be cited, for examples https://doi.org/10.1016/j.compbiomed.2022.105367

Response: Thank you. We have re-edited references accordingly. And we also cited more recent work on Omicron.

  1. Conclusion section should be more comprehensive and some future steps to fight this variant may be added.

Response: Thank you. We have revised the discussion section extensively.

  1. How much dangerous these variants need to be discuss (fatal rate), even though the transimissibility is high.

Response: Thank you. We have added the content about the disease severity caused by the SARS-CoV-2 variant infection.

Round 2

Reviewer 1 Report

Acceptable